# Effect of Rotational Modes on Torque/Force Generation and Canal Centering Ability during Rotary Root Canal Instrumentation with Differently Heat-Treated Nickel–Titanium Instruments

**DOI:** 10.3390/ma15196850

**Published:** 2022-10-02

**Authors:** Satoshi Omori, Arata Ebihara, Keiko Hirano, Yuka Kasuga, Hayate Unno, Taro Nakatsukasa, Shunsuke Kimura, Keiichiro Maki, Takao Hanawa, Takashi Okiji

**Affiliations:** 1Department of Pulp Biology and Endodontics, Division of Oral Health Sciences, Graduate School of Medical and Dental Sciences, Tokyo Medical and Dental University (TMDU), 1-5-45 Yushima, Bunkyo-ku, Tokyo 113-8549, Japan; 2Department of Metallic Biomaterials, Institute of Biomaterials and Bioengineering, Tokyo Medical and Dental University (TMDU), 2-3-10 Kanda-Surugadai, Chiyoda-ku, Tokyo 101-0062, Japan

**Keywords:** canal centering ability, heat treatment, nickel–titanium rotary instruments, optimum torque reverse motion, reciprocation, screw-in force, torque

## Abstract

This study aimed to evaluate how various rotational modes influence the torque/force production and shaping ability of ProTaper Universal (PTU; non-heat-treated) and ProTaper Gold (PTG; heat-treated) nickel–titanium instruments. J-shaped resin canals were instrumented with PTU or PTG using an automated instrumentation device operated with reciprocating rotation [150° clockwise and 30° counterclockwise (R150/30) or 240° clockwise and 120° counterclockwise (R240/120)], optimum torque reverse motion (OTR), or continuous rotation (CR) (n = 10 each). Maximum force and torque were recorded, and canal centering ratios were calculated. Statistical analysis was performed with two-way ANOVA and a Bonferroni test (α = 0.05). The results were considered with reference to previous studies on the microstructure of the instruments. The upward force generated by R240/120 and OTR was smaller than that generated by R150/30 and CR in PTG (*p* < 0.05). The clockwise torque produced by OTR was lower than that produced by R150/30 in PTU and R240/120 and CR in PTG (*p* < 0.05). R240/120 and OTR induced less canal deviation compared to CR in PTU at 0 mm from the apex (*p* < 0.05). In conclusion, R240/120 and OTR reduced the screw-in force in PTG and improved the canal centering ability in PTU, which may be associated with the heat treatment-induced microstructural difference of the two instruments.

## 1. Introduction

Nickel–titanium (NiTi) rotary instrumentation is progressively gaining popularity because of the superelastic properties of NiTi alloy that allow better preservation of the original root canal curvature [1,2]. However, intracanal instrument fracture remains a concern, and there is an ongoing trend toward its prevention, including improvements in the instrument design [3,4], kinematics [5,6], and metallurgy [7,8]. Among these, reciprocating kinematics with combinations of clockwise and counterclockwise rotation favor greater cyclic fatigue resistance and reduced torque development of NiTi instruments compared with continuous rotation, due to periodical relief of the stress [5,9]. Heat treatment of NiTi alloys is proven to be effective in improving cyclic fatigue resistance [7,10,11], through changing the phase transformation temperature and the resulting formation of a ductile martensite phase and R-phase [7,10].

In reciprocating NiTi file systems such as WaveOne (Dentsply Maillefer, Baillagues, Switzerland) and Reciproc (VDW, Munich, Germany), the clockwise and counterclockwise rotational motions are repeated at regular rotation angles in increments of 120°, and a file completes a 360° of rotation after three clockwise–counterclockwise cycles. Instruments used with this type of motion show higher cyclic fatigue resistance [9] and lower torque/force generation [5,12] compared with those used with continuous rotation. Regarding canal shaping ability, the reciprocating file systems in general show less canal transportation compared with conventional continuous rotating instruments [9,13]. Moreover, the reciprocal motion was reported to induce less canal transportation than continuous rotation when the same conventional instrument was tested with the two rotational modes [13]. However, there are contradictory findings, showing that reciprocating instruments exhibit similar or more canal transportation compared with continuous rotating instruments [9,14].

Recently, torque-sensitive reciprocal rotation—in which the motor changes its motion from continuous rotation to a reciprocal movement only when the torque exceeds the preset value—has been developed, aimed at reducing the instrument fatigue without decreasing the cutting efficiency [15]. In the optimum torque reverse (OTR) motion installed on the TriAuto ZX2 motor (J Morita, Kyoto, Japan), the clockwise and counterclockwise rotational angles are set at 180° and 90°, respectively [6,16]. The OTR motion is reported to improve cyclic fatigue resistance [16,17], generate less vertical force and torque [5,6], and show a similar [15] or slightly better [18] canal centering ability compared with continuous rotation.

Heat treatment of NiTi rotary instruments changes the phase transformation temperature of the NiTi alloy to induce crystal structure changes from the austenitic phase to martensitic phase and R-phase, which are softer and more ductile than austenite, at the clinical temperature [7]. The phase transformation leads to the formation of Ti_3_Ni_4_ precipitates finely dispersed in the austenite matrix phase and the formation of a titanium oxide layer on the surface. Therefore, the mechanical properties of NiTi rotary instruments are affected with or without heat treatment [19,20]. Many studies have shown improved cyclic fatigue resistance [10,19,21,22,23,24], increased flexibility [[10],[19][23],[24]], and improved shaping ability [11] following heat treatment. However, heat treatment is reported to reduce torsional resistance [10,20], and microhardness [20,23].

There is room for debate as to which rotational modes with what rotational angles are optimal for safe and efficient NiTi rotary instrumentation. Previous studies have shown that decreasing the reciprocation range in cutting direction (angle of progression in each reciprocation cycle) is a significant factor that increases the resistance to cyclic fatigue [25,26,27] and reduces canal transportation [26]. In contrast, limited knowledge exists regarding the effect of different reciprocation angles on the torque/force generation during instrumentation, although OTR motion is reported to generate similar torque and force compared with reciprocation with the same angles [5]. Moreover, no studies have been conducted regarding the performance of reciprocal movement with different reciprocal angles in relation to different heat treatment of NiTi instruments.

Therefore, the aim of this study was to examine the effect of various rotational modes with different rotational angles on the torque/force production and shaping ability of NiTi instruments of the same geometry with or without heat treatment. To our knowledge, this is the first study that examined the sole effect of metallurgy on the performance of different reciprocal movements, while eliminating confounding geometric factors. The null hypothesis was that there is no difference in the vertical force, torque, and canal shaping ability between the two types of instruments in different rotational modes. The results were considered with reference to the microstructures of the alloys observed by previous studies.

## 2. Materials and Methods

### 2.1. Sample Size Estimation

On the basis of a previous study [8], an effect size of 0.4, alpha-type error of 0.05, and study power of 0.8 were considered for the sample size calculation using G*Power 3.1.9 (Heinrich-Heine-Universität Düsseldorf, Düsseldorf, Germany). The sample size was estimated as ≥10 in each group.

### 2.2. Root Canal Instrumentation

ProTaper Universal (PTU) F2 (Dentsply Maillefer; made from non-heat-treated wire), and ProTaper Gold (PTG) F2 (Dentsply Maillefer; made from heat-treated Gold wire) were investigated. Both had the same geometrical features with a tip size of #25 and a 0.08 taper at the tip.

Eighty J-shaped transparent root canal models (Endo Training Bloc; 0.02 taper, 0.15 mm apical foramen diameter, 45° curvature, 16 mm full working length; Dentsply Maillefer) were used. Following coronal flaring to 12 mm from the orifice using PTG SX (Dentsply Maillefer) and manual glide path preparation with K-files (#10 and #15; Zipperer, Munich, Germany), the canals were sequentially instrumented to the full working length with PTG S1 (#18/0.02 taper) and S2 (#20/0.04 taper) files attached to the TriAuto mini endodontic motor handpiece (J Morita), followed by PTG F1 (#20/0.07 taper) files attached to an automated canal shaping device as described below. The canals were divided randomly into four groups as follows: (i) reciprocating rotation of 150° clockwise and 30° counterclockwise (R150/30); (ii) reciprocating rotation of 240° clockwise and 120° counterclockwise (R240/120); (iii) OTR motion at a torque value of 0.4 N·cm; (iv) continuous rotation (CR: no torque reverse). Each group was subdivided for PTU and PTG (n = 10 each).

Each canal was instrumented with PTU F2 or PTG F2 to the full working length using an automated canal shaping device, which consisted of a testing stand with a mobile stage, to which an endodontic handpiece modified from the TriAuto mini was attached. The stage made a repeated up-and-down movement at 20 mm/min for 2 s downward and 1 s upward [5,28]. The rotational speed of the handpiece was set at 300 rpm. The canal was lubricated with RC-Prep (Premier Dental, Plymouth Meeting, PA), and irrigated with 1 mL of distilled water once instrumented to 15 mm, followed by patency verification with a #10 K-file and instrumentation to the full working length. 

### 2.3. Force and Torque Analysis

The maximum vertical force (downward and upward) and torque (clockwise and counterclockwise) were determined using a torque/force analyzing device connected to the canal model as described previously [5]. Briefly, a load cell (LUX-B-ID, Kyowa, Tokyo, Japan) and strain gauges (KFG-2-120-D31-11, Kyowa) were used to detect vertical force and torque, respectively.

### 2.4. Canal Centering Ratio

Images of the canals were taken before and after instrumentation using a digital microscope (VH-8000; Keyence, Osaka, Japan; ×20 magnification), followed by superimposition and analysis using image analyzing software (Photoshop 7.0; Adobe Systems, San Jose, CA) as described previously [28]. The canal centering ratio was determined at 0, 0.5, 1, 2 and 3 mm from the root apex using the following formula: (X − Y)/Z, where X is the amount of material removed from the outer canal, Y is the amount of material removed from the inner canal, and Z is the postoperative canal diameter [28].

### 2.5. Statistical Analysis

Following verification of normal distribution and homogeneous variance using the Shapiro–Wilk test and Levene′s F test, respectively, two-way analysis of variance was performed, and the main effect and interaction were analyzed. If the interaction was significant, a simple main effect test was performed, and the Bonferroni test was used for multiple comparison (α = 0.05).

## 3. Results

### 3.1. Force and Torque Analysis

Downward force values among the four rotational modes were not significantly different in either instrument (*p* > 0.05; Figure 1A). For OTR and CR, PTG showed significantly lower downward force than PTU (*p* < 0.05; Figure 1A).

The upward force, representing the screw-in force [5,6,11,28], was smaller in R240/120 and OTR than in R150/30 and CR, when PTG was used (*p* < 0.05; Figure 1B). The upward force of PTU was smaller than that of PTG in all the rotational modes (*p* < 0.05; Figure 1B).

The clockwise torque produced by OTR was lower than that of R150/30 in PTU, and lower than that of R240/120 and CR in PTG (*p* < 0.05; Figure 1C). The clockwise torque of PTU was significantly lower than that of PTG in R240/120, OTR, and CR (*p* < 0.05; Figure 1C). 

Counterclockwise torque values were ranked as R240/120 > OTR > R150/30 > CR in both PTU and PTG (*p* < 0.05; Figure 1D). In R150/30, R240/120, and OTR, the counterclockwise torque of PTG was lower than that of PTU (*p* < 0.05; Figure 1D).

Figure 2 shows the representative torque/force recordings during PTG instrumentation. In R150/30 (Figure 2A) and R240/120 (Figure 2C), downward force and torque in both directions initially increased periodically, followed by sharp increases in clockwise torque and concomitant increases in upward force. Time-expanded diagrams near the maximum clockwise torque peak revealed that, in R240/120 (Figure 2D), intervals of the counterclockwise torque peaks were larger and accompanied by smaller increases in upward force compared with those of R150/30 (Figure 2B). In OTR (Figure 2E,F), upward force began to increase almost simultaneously with the increase in clockwise torque, followed by a sharp decrease in clockwise torque and upward force, and then torque and force developed intermittently in both directions. In CR (Figure 2G,H), torque and upward force increased rapidly at the same time.

### 3.2. Canal Centering Ratio

Significant differences among rotational modes were noted in PTU at the 0 mm level, where R240/120 and OTR recorded lower values (less deviation) than CR (*p* < 0.05; Table 1). PTG recorded a significantly lower centering ratio than PTU at one or several measuring levels in each rotational mode (*p* < 0.05; Table 1).

## 4. Discussion

This study attempted for the first time to evaluate torque/force generation and the canal shaping ability of NiTi rotary instrumentation with various rotational modes using different heat-treated instruments. The main findings were that R240/120 and OTR significantly reduced upward force in PTG and canal centering ratios in PTU, when compared with CR. From these significant differences, the null hypothesis was rejected.

In this study, simulated resin root canals were employed to exclude the differences in root canal anatomy in natural teeth, thereby standardizing the experimental conditions [29]. However, resin canals are softer than dentin and are prone to clogging by cutting fragments, which might cause a sudden increase in torque [29,30]. Further evaluation is required using natural teeth with as little individual variation as possible. 

PTU and PTG were chosen to eliminate confounding factors other than metallurgy. Owing to the heat treatment, PTG has a two-stage transformation behavior [19], and the austenite finishing temperatures of PTU and PTG are reported to be 21.2 ± 1.9 °C and 50.1 ± 1.7 °C, respectively [19]. Thus, under the present experimental conditions at room temperature, PTU was austenitic, whereas PTG was in a mixed phase containing martensite, austenite, and R-phase. PTG is reported to show higher flexibility [19] and less canal transportation [31] than PTU, which may be largely attributable to the heat treatment-induced crystallographic changes and explains the present finding that PTG in general recorded a lower centering ratio than PTU in each rotational mode. Regarding fracture resistance, PTG is reported to show higher cyclic fatigue resistance [19,21,23,32], but lower torsional resistance [20,21,33] than PTU. Both instruments exhibit a similar fracture surface ultrastructure, i.e., circular abrasion streaks and skewed dimples in the center of the fractured surfaces following torsional loading [20,23,33], as well as a similar crack initiation area and a fast fracture zone following cyclic loading [19,20]. Thus, the microstructural difference may not influence the basic mechanics of instrument fracture. However, further study is needed to determine how the differences in the metallurgical microstructure affect the performance of heat-treated and non-heat-treated instruments, particularly under different rotational modes, i.e., continuous rotation or reciprocation.

In reciprocal root canal instrumentation, reducing the reciprocation range by changing the rotational angle is reported to improve cyclic fatigue resistance and canal shaping ability [25,26,27]. However, few reports have compared reciprocating rotations that differ in rotational angle but share the same reciprocation range. In this study, R150/30 was adopted in reference to the Reciproc system, and R240/120 was employed to examine the effect of a larger reciprocal amplitude, with an assumption that the larger counterclockwise angle in R240/120 may effectively release the stress to the instruments and may be beneficial in reducing torque/force generation.

The screw-in force, detected as the upward force, pulls the instruments in an apical direction [5,6,11,28] and may cause momentary torsional stress, leading to separation of the instruments [34,35,36]. In PTG, OTR significantly reduced the screw-in force when compared with CR, supporting earlier findings [5,6,15]. Moreover, R240/120 generated a significantly smaller screw-in force than R150/30 during PTG instrumentation. This may be attributed to the larger counterclockwise rotational angle in R240/120, which may have promoted the disengagement of the blades. This assumption is supported by the torque/force recordings; in R240/120 (Figure 2D), larger intervals of counterclockwise torque peaks, resulting from the larger counterclockwise rotational angle, were accompanied by smaller increases in the screw-in force when compared with R150/30 (Figure 2B).

PTG yielded a significantly larger screw-in force than PTU in all rotational modes. This may be associated with the higher cutting efficiency of PTG than PTU [37], which could be because the flexibility of PTG allowed more evenly distributed contact of cutting edges with the canal wall [37]. This could promote more pronounced threading, resulting in screw-in force generation. However, conflicting results, probably due to differences in the mode of up-and-down motion, have been reported [12], and further studies are warranted.

This study failed to detect any significant differences in the maximum downward force among groups, which supports the finding that CR and OTR developed similar downward force [6]. PTG developed significantly smaller downward force than did PTU in OTR and CR, probably resulting from the higher screw-in tendency of PTG, via which the downward force was released in the opposite direction. 

OTR developed significantly smaller clockwise torque than R240/120 in PTG and R150/30 and CR in PTU. The recordings of OTR (Figure 2E,F) showed a sharp decrease in increased torque and upward force, indicating quick stress release. These findings indicate that torque-sensitive reciprocation is more effective in disengaging a file from the canal wall than a time-dependent reciprocating motion. The counterclockwise torque values were R240/120 > OTR > R150/30, indicating that a larger counterclockwise rotational angle is associated with a larger counterclockwise torque. PTG showed a smaller counterclockwise torque than PTU, in accordance with the finding that flexible instruments generate lower counterclockwise torque [24].

R240/120 and OTR yielded less canal deviation than R150/30 and CR at the apex, which is consistent with the finding that reciprocation is superior to CR in reducing canal straightening [38]. This is possibly because larger counterclockwise rotational angles more effectively disengage the instrument from the canal wall, thereby reducing the blade–canal wall contact time at the outer canal wall [28]. Such an effect of R240/120 and OTR was noted only in PTU, indicating that the flexibility of the instrument is more decisive in determining the degree of canal straightening than the rotational mode.

The results from the present study suggest the clinical advantage of reciprocal movements, particularly OTR, in terms of (i) reducing torsional stress due to torque and screw-in force generation in heat-treated NiTi instruments, and (ii) compensating for the tendency of non-heat-treated instruments to induce canal deviation. However, further research is needed to determine optimal rotational modes according to different types of NiTi rotary instruments with different metallurgy and geometry.

## 5. Conclusions

Within the parameters of this study, it can be concluded that OTR contributed to reducing the screw-in force in PTG and improving the canal centering ability in PTU, when compared with continuous rotation. This may be associated with the heat treatment-induced microstructural difference of the two instruments. R240/120, but not R150/30, exhibited a similar tendency, indicating that reciprocation with a larger reciprocal amplitude could provide advantages similar to those of OTR, when motions with the same reciprocal range were compared.

## Figures and Tables

**Figure 1 materials-15-06850-f001:**
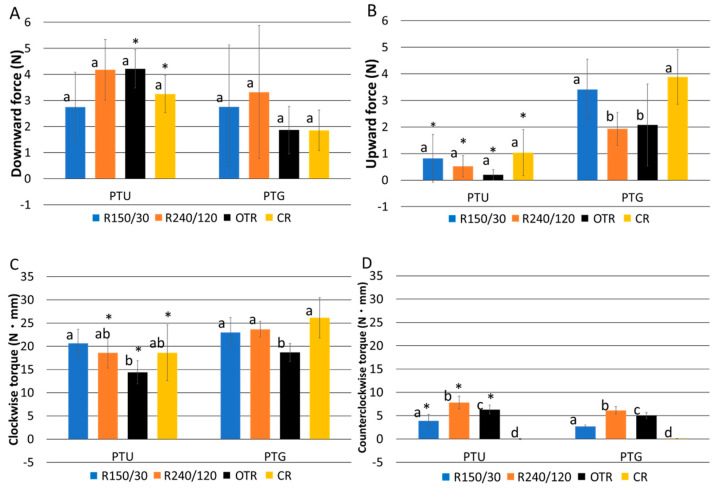
The maximum values of (**A**) downward force, (**B**) upward force, (**C**) clockwise torque, and (**D**) counterclockwise torque during root canal instrumentation. Data represent the mean and standard deviation (n = 10). Different letters indicate a significant difference (*p* < 0.05) between groups within the same instrument. Asterisks indicate a significant difference compared with corresponding values for the PTG group (*p* < 0.05).

**Figure 2 materials-15-06850-f002:**
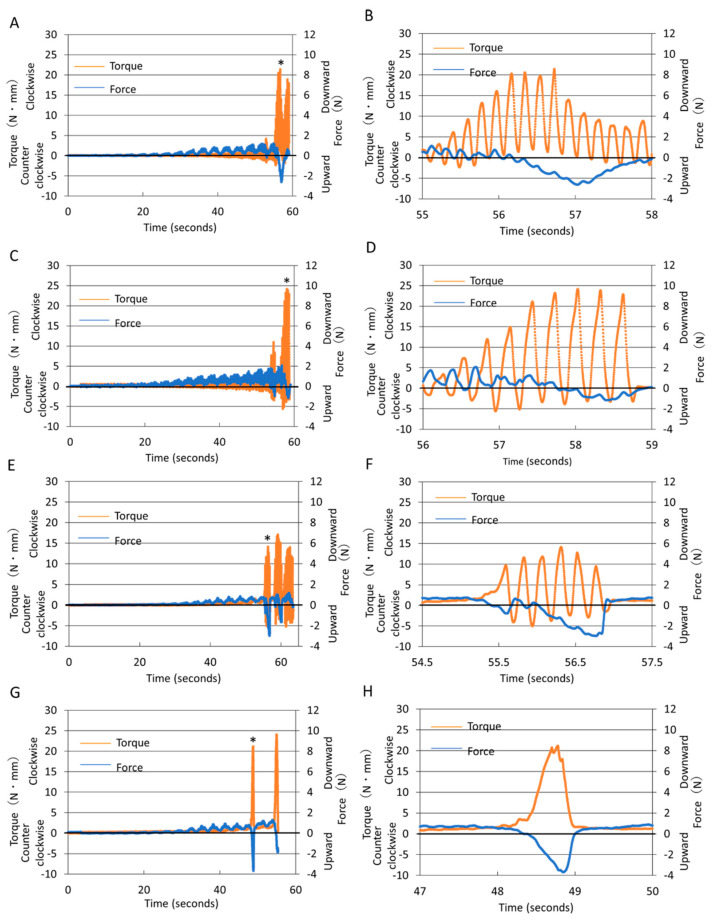
Representative recordings of torque and force values during instrumentation with R150/30 (**A**,**B**), R240/120 (**C**,**D**), OTR (**E**,**F**), and CR (**G**,**H**). (**B**,**D**,**F**,**H**) Time-expanded diagrams near the maximum clockwise torque peak (*) in (**A**,**C**,**E**,**G**). Positive and negative force and torque values refer to upward and downward force and clockwise and counterclockwise torque, respectively. Labels on the x-axis indicate time (seconds) from the initiation of instrumentation.

**Table 1 materials-15-06850-t001:** Canal centering ratios.

Level from Apex (mm)	Instrument	Rotational Mode			
		R150/30	R240/120	OTR	CR
0	PTU	0.31 ± 0.10 ^ac,A^	0.17 ± 0.09 ^b,A^	0.22 ± 0.07 ^bc,A^	0.34 ± 0.07 ^a,A^
	PTG	0.15 ± 0.06 ^a,B^	0.17 ± 0.06 ^a,A^	0.18 ± 0.08 ^a,A^	0.22 ± 0.09 ^a,B^
0.5	PTU	0.21 ± 0.07 ^a,A^	0.15 ± 0.06 ^a,A^	0.21± 0.10 ^a,A^	0.20 ± 0.06 ^a,A^
	PTG	0.09 ± 0.05 ^a,B^	0.13 ± 0.07 ^a,A^	0.14 ± 0.05 ^a,^^B^	0.17 ± 0.05 ^a,^^A^
1	PTU	0.23 ± 0.08 ^a,A^	0.15 ± 0.07 ^a,A^	0.17 ± 0.07 ^a,A^	0.20 ± 0.05 ^a,A^
	PTG	0.10 ± 0.04 ^a,B^	0.14 ± 0.06 ^a,A^	0.16 ± 0.06 ^a,A^	0.17 ± 0.06 ^a,A^
2	PTU	0.20 ± 0.06 ^a,A^	0.19 ± 0.07 ^a,A^	0.18 ± 0.09 ^a,A^	0.24 ± 0.04 ^a,A^
	PTG	0.13 ± 0.03 ^a,B^	0.12 ± 0.04 ^a,B^	0.16 ± 0.08 ^a,A^	0.17 ± 0.05 ^a,B^
3	PTU	0.11 ± 0.08 ^a,A^	0.10 ± 0.06 ^a,A^	0.13 ± 0.06 ^a,A^	0.09 ± 0.05 ^a,A^
	PTG	0.03 ± 0.01 ^a,A^	0.04 ± 0.03 ^a,A^	0.04 ± 0.02 ^a,B^	0.06 ± 0.03 ^a,A^

Data represent the mean and standard deviation (n = 10). Different uppercase letters indicate that the values are significantly different (*p* < 0.05) within the same rotational mode at the same measurement level. Different lowercase letters indicate that the values are significantly different (*p* < 0.05) within the same instrument.

## Data Availability

The data presented in this study are available on request from the corresponding author.

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
