# Peer review of "Effect of Rotational Modes on Torque/Force Generation and Canal Centering Ability during Rotary Root Canal Instrumentation with Differently Heat-Treated Nickel–Titanium Instruments"

_materials, 2022, doi:10.3390/ma15196850_

Round 1

Reviewer 1 Report

In this manuscript, the authors compared the effects of various rotational modes on the torque and force of two types NiTi instruments. This study has certain reference value for optimizing stress application or usage of NiTi root canals. However, from the perspective of its theme, this study lays more emphasis on the mechanical behavior and mechanical properties of a instruments than on the material issues. So, I think it is more appropriate for this manuscript to be transferred to a journal in machinery field.

Reviewer 2 Report

The authors studied the effect of rotational Modes on torque/force generation and 2 canal centering ability during rotary root canal instrumentation with differently heat-treated Nickel-Titanium Instruments. The research work is good and suitable for publication. However, following corrections must be done before publication.

1. The authors should discuss the research work in introduction section instead of giving citations.

2. The quality of figures need to be improved.

Reviewer 3 Report

The paper presents interesting information on the performance of root canal rotary instruments. The following points should be considered during revision.

 1. Novelty should made clear at the end of introduction.

2. Section 2.5 equation should be written with math type form with an equation number

3. Table 2 notes. Confirm different small or capital letters indicate significance both along the rows and columns.

4. Discuss further limitations and clinical significance of this study.

5. Even though you have used two different heat-treated instruments but the results were not connected with their metallurgical structure. This can be supported by SEM  images of microstructures.

Round 2

Reviewer 1 Report

In the revision, the authors gave a careful reply to my unfriendly comments and revised the manuscript according to them. I think these modifications improve the “material” flavor of the manuscript. Additionally, I also read the modifications to the other reviewer, which significantly improved the quality of the manuscript. Therefore, I think the manuscript in this version can be published.